# Trends in Access to Oral Health Care among Adults from the N-E Region of Romania

**DOI:** 10.3390/medicina59010074

**Published:** 2022-12-29

**Authors:** Walid Edlibi Al Hage, Cristina Gena Dascălu, Carina Balcoș, Doriana Agop-Forna, Norina Consuela Forna

**Affiliations:** 1Department of Implantology, Removable Prostheses and Dental Prostheses Technology, Faculty of Dental Medicine, “Grigore T. Popa” University of Medicine and Pharmacy, 700115 Iasi, Romania; 2Informatics Department, Faculty of Medicine, “Grigore T. Popa” University of Medicine and Pharmacy, 700115 Iasi, Romania; 3Department Surgery, Faculty of Dental Medicine, “Grigore T. Popa” University of Medicine and Pharmacy, 700115 Iasi, Romania

**Keywords:** access to dental care, adults, income, questionnaire

## Abstract

*Aims and Objectives*: To investigate the trends in access to dental services among adults from the N-E region of Romania and to evaluate the factors that influence access to dental care. *Material and Methods*: A self-administered questionnaire was used to evaluate the access and utilization of dental care among a sample of adults. We collected data on their demographic characteristics, their type of medical insurance, their monthly income, the type of dental office they visited, their reasons for choosing a certain type of dentist office, and their oral hygiene habits. It was found that their financial status determined by the occupation of the individual, as well as their monthly income, influenced their access to medical services. Data were analyzed using SPSS 20.0. *Results*: Of a total of 696 participants, 55.6% were female, 83.3% were from urban areas, 42.1% were retired, 62.3% of the subjects utilized dental emergency services, and 67.9% of the subjects self-funded their treatment. The reasons for women not attending dental offices included costs (24.3%) and dental fear (9.8%), while men’s reasons were high costs (26.4%) and lack of time (5.9%). Significant differences were recorded between gender and reasons for visiting the dentist (*p* = 0.018), payment for dental services (*p* = 0.009), and preferred clinic (*p* = 0.010). In relation to occupation, there were significant differences for most of the variables evaluated (reasons to visit a dentist, payment for dental services, preferred dental clinic, self-rated oral health, etc.). *Conclusions*: Gender, occupation, and monthly income levels were found to influence access to dental medicine services of Romanians in the N-E region. Dental services were frequently accessed for emergency reasons. Dental services were found to be paid for by state insurance for those with low monthly incomes and self-funded for those with higher monthly incomes.

## 1. Introduction

Oral health is an integral part of general health and well-being, as stipulated by the multitude of definitions that already exist in specialized literature [1,2]. Although all specialists agree with this definition and given that all the efforts made to reduce the number of people with oral diseases globally were significant, the number of individuals with chronic oral diseases remains high (over 3.5 billion estimated in 2019) [3] due to the persistence of risk factors involved in their occurrence. Factors such as socio-economic inequality, a lack of health education, cultural habits, and a lack of access to medical services continue to implicitly influence oral health and the quality of life of everyone, especially those in developing countries [4].

The prevalence of chronic oral diseases among the young and adult population in Romania is high, although the number of studies carried out in Romania is small. The results of these studies show us that the prevalence of dental caries in the N-E region of Romania was 96.3% in young adults and 97.5% in the adult population, with women being more affected than men [5]. The results of the study carried out by Forna N. et al. in which the oral health of the adult population was analyzed over a period of 10 years (2009–2019) indicate that the prevalence of dental caries was higher in rural areas compared to urban areas [6]. Chronic diseases such as oral cancer in Romania have high prevalence and incidence rates, a situation confirmed by the latest World Health Organization (WHO, Geneva, Switzerland) data published in 2020, with deaths caused by oral cancer in Romania reaching 2290 (or 0.98% of the total deaths recorded), i.e., the 16th highest of all countries around the world [7]. The late detection of oral diseases is determined by a lack of knowledge regarding the symptoms of these diseases and by the low financial power.

The population’s access to medical treatments is hindered by various factors such as the type of health system that each state has, the culture and education of the population, as well as the income [8]. Organized according to various models, each country’s health system represents a determinant of health. Given the preventive character of the health system in Nordic countries, most funds are invested in programs in Eastern Europe for the prevention of oral diseases and for periodic check-ups [9], as well as for the treatment of these diseases [10,11]. The reforms introduced over the past few years have incorporated a range of health benefits provided by Romania’s National Health Insurance House (NHIH) and levels of public spending on health have increased; however, still, the allocated funds are extremely low, i.e., only 5.7% of people are covered by general medical insurance. Therefore, access to dental care is limited due to budget constraints [12,13]. Due to a lack of infrastructure and primary healthcare facilities, access to healthcare is especially limited in rural areas, which is exacerbated by significant gaps in health insurance coverage. Even though many people are exempt from paying health insurance contributions (including children and people with disabilities), the number of people with health insurance is decreasing year after year [12].

Another aspect that influences access to medical services supported by the government is the fact that the dental medicine system is 99% private, consisting of private dental practices that can offer dental medical services compensated by the state through Romania’s National Health Insurance House (NHIH, Bucharest, Romania), with the remaining 1% being represented by the faculties of dental medicine or emergency dental offices within university hospitals that offer compensated dental services. The amount granted monthly by a doctor differs depending on whether the doctor is a specialist or not, or if he or she works in a rural or urban area. This amount does not exceed EUR 900 per month for a specialist and can cover the treatment for a maximum of 1–2 patients a month who require complex (prosthetic) treatments; this situation thus further restricts access to medical services. The economic crisis induced by the COVID-19 pandemic as well as the existing war in the region increases the pressure on the population that already has deficiencies in accessing services for the previously stated reasons [14].

Another obstacle to accessing medical services is the payment method of for services. Patients with a high socio-economic level prefer to pay for dental medical services from their own funds (out-of-pocket payments) for dental check-ups, as well as for curative treatments in private clinics. Patients with a low socio-economic status generally only visit the dentist when required (e.g., for emergencies) [15].

Not all regions of Romania have the same socio-economic level; for instance, regions in the south and west have better ratings than the N-E regions (Moldavia) which have always had a much lower registered GDP compared to the other regions of Romania. This corroborates the action of other factors that influence access to medical services [16,17]. All these listed factors influence the hygiene behavior of inhabitants of various regions of the country; thus, our study aimed to highlight the factors that influence the access of adult residents from the N-E region of Romania to dental medical services. The working hypothesis is that the gender, occupation, and monthly income levels of participants significantly influence access to medical services, as well as hygiene attitudes.

## 2. Materials and Methods

This retrospective observational study was conducted among the adult population from the N-E region of Romania, after obtaining the approval of the Ethics Commission of the University of Medicine and Pharmacy”, Grigore T. Popa Iasi, Romania (no. 231/13.10.2022). Data on the factors that influence access to specialized medical services were collected through questionnaires distributed to the participants in an oral health screening action called “Prophylaxis Caravan”, an action that has taken place annually in the N-E region of Romania over the last 10 years with the aim of screening oral diseases among the population of the N-E region of Romania. The data were selected from the questionnaires collected in the year 2022.

Participants were selected if they were: over 18 years of age, had permanent residence in the N-E region of Romania, signed the informed consent and completed the full questionnaire, and received a briefing on what the study consisted of under conditions of anonymity. To an adult population of 3.712.396 people [18], we applied the calculation formula with a confidence level of *p* = 95%, z = 1.96, as well as a margin of error of 5%. The sample size was 385 adults [19]. Of the total questionnaires completed, 696 were completed by adults, which is why we selected all the questionnaires completed by adults in 2022.

In addition to questions relating to demographic characteristics (age, sex, residence, occupation), the questionnaire contained questions relating to the participants’ type of medical insurance, their monthly income, their type of dental office, their reasons for choosing a certain type of dental office, as well as their oral hygiene habits (the number of daily tooth brushings, toothbrush rotation, and the frequency of dental check-ups). The financial status determined by the occupation of the individual as well as their monthly income influences access to medical services.

The data collected were analyzed using SPSS Software (Version 20.0) (SPSS^®^ Inc., Chicago, IL, USA) to generate descriptive statistics and analyze the data. Descriptive statistics were presented as frequency, percentages, means, and standard deviations. Age, gender, occupation, monthly income, and the number of visits to the dentist were considered the most important variables that can describe the attitude of adults regarding access to dental services. Pearson’s chi-square test was used to find an association between categorical variables. A ***p***-value less than 0.05 was considered statistically significant.

## 3. Results

Table 1 shows the analysis results regarding the study group’s demographic data. Thus, of the 696 participants in total, 55.6% were female subjects, 83.3% came from urban environments, and 42.1% were retired. The average age was 49.66 ± 18.21 years (min. of 19 years and max. of 85 years) (Table 1).

From a financial point of view, 43.5% of the participants were found to have a monthly income between EUR 501 and 1000 and 67.9% were self-financed, 25% of whom had state insurance. More than 70% of the participants preferred to access private dental services and only 20% favored the state ones.

An evaluation of the general health status demonstrates that 42.7% of the participants had systemic diseases. A self-assessment of the state of health indicates that 52.4% perceived their general state of health as “very good” and 27% as “good”. Regarding the need for dental treatment, 60.5% of the participants considered that they do not need dental treatment (Table 1).

The evaluated hygiene habits indicate that 58.1% of the subjects brush their teeth twice a day and 59.9% change their toothbrush 1–3 times/per year. Regarding the number of visits to the dentist, 33.7% visit the dentist “when needed” and 33.5% do so “once a year” (Table 1).

Access to dental services was shown to vary depending on various factors. Thus, male subjects declared that they visit the dentist once a year (33.5%) and women do so when required (33.7%). Participants from rural areas tend to visit the doctor when they had to, while 34.9% of participants from urban areas visit the doctor’s office at least once a year. Employed subjects access dental services at least twice a year, while unemployed or retired subjects visit the dentist when needed. The recorded differences were statistically significant (*p* = 0.000) (Table 2).

The subjects with a monthly income > EUR 500 declared that they visit a doctor at least once a year and those with EUR < 500 do so only when needed (e.g., for medical emergencies). For preventive checks, participants visit the doctor once (58.9%) or twice/year (38.9%). In total, 62.3% of the subjects declared that they would only visit the emergency room when they needed to, and only 34.7% would so for specific treatments. The differences were statistically significant for the “occupation” variables, the “reasons for presenting to the dentist”, and the “payment method” for medical services (Table 2).

Payments of medical services were found to be made mainly from own sources, especially when the subjects call on the services of private dental offices. Those who benefited from government-settled services came to the dental office more “as needed” (41.9%) (Table 2).

Table 3 shows the relationship between the gender and monthly income variables and the factors influencing access to dental services. Female subjects said that they brush between two and three times a day (60.4% and 68%, respectively) and change their brushes annually more frequently (72.5%) (*p* = 0.001, *p* = 0.000). Moreover, they had more annual treatment sessions than male subjects, possibly due to the preventive control and treatment of dental problems, for which they paid more from their own sources. Significant differences were recorded in the case of “reasons to visit the dentist” (*p* = 0.018), “payment for dental services” (*p* = 0.009), and “preferred dental clinic” (*p* = 0.010) (Table 3).

Women did not regularly visit the doctor’s office due to “costs” (24.3%) and ”fear of dental procedures” (9.8%), while men’s reasons were “high costs” (26.4%) and “lack of time” (5.9%).

Female subjects perceived their own oral health status as “very good” (55.1%) and “good” (29%), which was higher compared to male subjects. Male subjects, on the other hand, were found to acknowledge the necessity of treatment as being higher than female subjects.

Regarding the relationship between healthy habits and monthly income, the results indicate that, on a declarative level, more than half of the subjects admitted to brushing their teeth two times a day and changing their toothbrushes between one and three times a year. Those with incomes below EUR 500 per month had a percentage higher than those who brushed from time to time and changed their toothbrushes as needed. Those with incomes below EUR 500 said to visit the doctor “when needed” (40%), such as for medical emergencies (47.9%) or if expenses were covered by state medical insurance (88%), and those with incomes more than EUR 1000 proclaimed to have 1–2 annual sessions (32%), and only increased in the case of emergencies (35%) and if they had to fund the service themselves (94% and 85.7%, respectively). These differences are significant (*p* = 0.000) between groups in relation to the financing of medical services.

An assessment of the data on the occupations of participants indicates that students perform two brushings every day in a higher proportion compared to the other categories. Those employed change their toothbrush at most three times a year and mostly visit the dentist for check-ups. These treatments and check-ups were found to be carried out in private clinics (77.2%) and are usually self-funded (85%). Retirees appeared to have good hygiene habits (brushing two times a day, changing up to three brushes a year), but they only visit the dentist for medical emergencies, and medical payments were found to be made from both state insurance and from their own pockets in almost equal proportions (49.3% and 48.3%, respectively) (Table 3).

The multivariate logistic regression (Table 4) shows us that in the case of the number of tooth brushings, male subjects have a greater predisposition to perform more brushings per day than female subjects (*p* = 0.001, OR = 1.76), the same positive trend being recorded and in the case of those employed (*p* = 0.000, OR = 2.14). 

In the case of changing the toothbrush, significant differences are observed in terms of the distribution of the results by gender, with female subjects being 2.714 more predisposed to change their toothbrush more than 3 times/year by compared to male subjects (*p* = 0.000, OR = 2.714), the same positive trend being registered in the case of the variable “occupation” where the employed will change their brushes more than 3 times a year 1.57 more than the unemployed subjects (*p* = 0.031, OR= 1.57), the difference being statistically significant (*p* = 0.031) (Table 4).

In the case of “self-rated oral health”, statistically significant differences were recorded in the case of the gender and occupation variables (*p* = 0.035, *p* = 0.000 respectively). Male subjects and the unemployed have a greater tendency to evaluate their own oral health as fair or poor (*p* = 0.000, OR = 0.493) (Table 4).

## 4. Discussion

Oral health, a major public health problem for any country, regardless of the level of development due to increased treatment costs, represents an indicator of the living and education levels of individuals [20,21,22]. Both socio-economic and cultural determinants influence the oral health of Romanians who, although they benefit from the support of the medical insurance system and the increased number of dentists, have poor oral health [23,24,25,26,27].

In Romania, oral health services are provided by private and state clinical centers in urban and rural areas that are financially supported by the state or personal sources (private insurance or out-of-pocket payments). A big impediment in accessing the services supported by state insurance is the type of medical treatment settled in Romania for adults, within the limit of 60% of costs; this is only restricted to odontal, periodontal, endodontic, and minimal prosthetic treatments (one removable acrylic prosthesis every 10 years or metal acrylic crowns), or dental extractions [23,28].

The hypothesis that the gender, occupation, and monthly income levels of the participants can influence access to medical services, as well as their hygiene attitudes, is supported by the results obtained in our study.

Studies confirm that low individual incomes are associated with oral cancer, increased dental caries prevalence, caries experiences, tooth loss, traumatic dental injuries, periodontal disease, and poor oral-health-related quality of life [29,30]. In our study, the incomes of the participants were modest, with less than half of them having incomes between EUR 501 and 1000 per month, thus significantly influencing their access to medical services, i.e., either general medicine or dentistry, especially given that 99% of dental services are private. The results of our study indicate that over 60% of those who came to the private dental office paid for their treatments from their own pocket. Many of the subjects with moderate monthly incomes visited the dentist at least once a year, and those with low income only did so when they had medical emergencies. This result confirms previous findings which indicate that socioeconomic conditions influence the usage of dental services [31,32,33].

Inequalities in access to medical services are found in all countries of the world, including all Europe countries. Eliminating financial barriers to healthcare access may have a positive effect on the utilization of oral healthcare [34,35,36,37] by supporting costs for dental treatments and prevention programs. This is because, according to 2015 data, the global cost of treating dental conditions for one year was EUR 442 billion, including both direct treatment costs and indirect costs caused by school and work absenteeism [38].

Regarding the relationship between oral hygiene attitudes and monthly incomes, the studies conducted in this field confirm the link between socioeconomic status, oral health in an individual, and various factors which directly or indirectly affect oral health [39]. Our results show that subjects with low incomes have deficient hygiene behavior, many of the subjects brush their teeth from time to time, and toothbrushes are changed “when needed”. In the case of participants with a high income, it was observed that the number of annual sessions is higher compared to those of subjects with lower incomes, but they mainly visited the doctor for medical emergencies. What is important to emphasize is that everyone perceived their own oral health as good without the need for dental treatment, i.e., a perception that can influence access to dental medical services. This behavior can be explained by the relatively modest level of oral education and low-income levels among adults in Romania, which results in poor oral health among them compared to other countries [5].

The method of paying for medical services is another element that influences the frequency of access to medical services. In our study, most participants were found to pay out-of-pocket expenses, followed by those who benefit from state medical insurance; this situation is due to the non-performing health insurance system in Romania. Thus, those with low incomes pay their medical expenses through state medical insurance, while those with moderate and high incomes finance their services from their own pockets. Thus, income levels can be a significant predictor of the non-utilization of dental services among adults. The results of our study agree with those of other studies in the literature too [40,41,42].

In countries where an insurance system is set up, patients will receive treatments compensated by state or private insurance, in state or private clinics. More than 70% of the participants in our study proclaimed to visit private dental offices, motivated by the fact that there are no longer many state dental clinics, but also by the idea that they will benefit from quality medical services. The studies carried out so far emphasize the preference for private offices because of the availability of different types of treatment, the quality of dental care, the ease and early availability of appointments, the short waiting time, and the possibility of continuing treatment. This finding is similar to the study reported by Obeidat et al. [12,41,43]

The low number of subjects who benefited from state-settled medical services is mainly due to the small budget provided by the Romanian state for dental services (approximately EUR 900 per month for a specialist doctor. Many private dental offices provide medical services that are not supported by the government because the budget is low and the bureaucracy is high [44,45].

Other factors such as employment, gender, and residence can influence access to medical services. Employed subjects visit the dentist at least twice a year, while unemployed or retired subjects go when needed due to low income levels and perhaps a lack of education. While the male subjects were found to visit a dentist once a year, it was found that the female participants only did so “when needed”. The participants from rural environments were found to only visit a doctor when they urgently need treatment, while those from urban environments do so at least once a year due to the higher income of the latter and because there are more offices in urban environments. In most cases, the behaviors of rural people can often delay access to health services because they believe that their oral health is good or simply because they do not have dental pain. The results obtained correspond with those of other studies in the literature [46,47,48]. The same observation was made by Al-Ansari, who found that the costs, the unavailability of doctors, and the lack of insurance were the main reasons for the reduced access to preventive oral health care [49,50,51].

Preventive check-ups and emergencies were found to be the main reasons for visiting the dentist. It was also discovered that less than 40% of the participants show up for treatments. The reasons for not regularly visiting the office are the costs and the fear of dental procedures, especially for women. For men, however, high costs and lack of time are the main reasons. The results of our study agree with those of another study conducted by Ajayi and Arigbede which identified the cost of dental treatment as a major barrier to oral healthcare utilization, but they observed a more significant association between access to care and fear dental treatment [49,52,53].

The limitations of our study can be attributed to the fact that the descriptive study was carried out with a self-reported questionnaire, accompanied by considerable subjectivity that can influence the analysis of the causal link between variables. Carrying out a longitudinal study on a larger number of participants can help to accurately establish all the factors that reduce access to dental medical services in Romania.

## 5. Conclusions

The results of our study indicate that access to the dental medicine services of Romanians in the N-E region is influenced by variables such as gender, occupation, and monthly income levels. Dental services are frequently accessed in the case of a dental emergency, more by female subjects and those from rural environments. Dental services are mostly paid for in private dental offices and are covered by state insurance for those with low monthly incomes, but are self-funded by those with a higher monthly income. Inequalities in accessing medical services in Romania can only be solved by increasing the funds allocated for dental services, as well as by establishing preventive programs to improve the level of education surrounding oral health among the adult population.

## Figures and Tables

**Table 1 medicina-59-00074-t001:** Distribution (%) of subjects according to sociodemographic, general, and oral-health-related variables (N = 696).

		No	%
Age	49.66 ± 18.21 years(min. 19, max. 85)		
Gender	Female	376	55.6
	Male	320	44.4
Residence	Urban	579	83.3
	Rural	117	16.7
Occupation	Student	46	6.7
	Employee	271	38.9
	Unemployed	86	12.3
	Retired	293	42.1
Monthly income	EUR <500	197	28.2
	EUR 501–1000	303	43.5
	EUR >1000	196	28.2
Payment for dental services	Never been to a dentist	14	2.0
	State insurance	174	25.0
	Private insurance	35	5.0
	Self-funded	473	67.9
Preferred dental clinic	I don’t frequent any clinic	14	2.0
	Private clinic	544	78.2
	Government clinic	138	19.8
Systemic health problems	Yes	297	42.7
	No	399	57.3
Self-rated oral health	Very good/good	365	52.4
	Fair	188	27.0
	Poor/very poor	143	20.6
Self-rated dental treatment need	Yes	275	39.5
	No	421	60.5
Oral hygiene habits			
Number of brushing/day	1 toothbrush /day	170	24.4
	2 toothbrushes/day	404	58.1
	3 toothbrushes/day	70	10.1
	From time to time	52	7.5
Number of toothbrushes/year	1–3 times/year	417	59.9
	4–6 times/year	184	26.4
	When needed	95	13.7
Number of visits to the dentist/ year	Less than once a year	33	4.8
	Once a year	233	33.5
	Twice a year	181	26.0
	When needed	235	33.7
	I didn’t go to the dentist	14	2.0

**Table 2 medicina-59-00074-t002:** Factors related to dental service’s frequency of utilization.

Variable		Number of Visits to the Dentist					*p*
		Less Than Once a Year	Once a Year	Twice a Year	When Needed	I Didn’t Go to the Dentist	
Gender	Female	4.3%	33.3%	26.4%	35.5%	0.4%	0.051
	Male	5.5%	33.6%	25.5%	31.4%	4.1%	
Residence	Urban	4.4%	33.2%	25.7%	34.9%	1.9%	0.648
	Rural	7.2%	34.9%	27.7%	27.7%	2.4%	
Occupation	Student	0.0%	36.4%	36.4%	27.3%	0.0%	0.000 *
	Employee	4.7%	34.2%	31.6%	24.9%	4.7%	
	Unemployed	8.2%	18.0%	32.8%	41.0%	0.0%	
	Retired	4.8%	36.8%	17.2%	40.7%	5.0%	
Monthly income	EUR < 500	4.3%	34.3%	20.0%	40.0%	1.4%	0.418
	EUR 501–1000	4.2%	33.8%	26.9%	33.3%	1.9%	
	EUR > 1000	6.4%	32.1%	30.7%	27.9%	2.9%	
Reasons to visit a dentist	Check-up	1.2%	58.9%	39.9%	0.0%	0.0%	0.000 *
	Emergency	7.0%	13.6%	16.6%	62.3%	0.5%	
	Treatment	5.6%	34.7%	25.0%	34.7%	0.0%	
	I didn’t go to the dentist	0.0%	0.0%	0.0%	0.0%	100%	
Payment for dental services	Never been to a dentist	0.0%	0.0%	0.0%	0.0%	100%	0.000 *
	State insurance	4.0%	37.9%	16.1%	41.9%	0.0%	
	Private insurance	4.0%	40.0%	12.0%	44.0%	0.0%	
	Self-funded	5.3%	32.3%	31.5%	30.9%	0.0%	

* Statistically significant differences when *p* < 0.05 (ANOVA test).

**Table 3 medicina-59-00074-t003:** Relationship between the gender and monthly income variables with factors that can influence access to dental services.

		Gender		Monthly Income			Occupation			
		Female	Male	EUR < 500	EUR 501–1000	EUR > 1000	Student	Employed	Unemployed	Retired
Number of toothbrushing/day	1 toothbrush /day	43.8%	56.2%	21.4%	24.1%	27.9%	3.0%	21.2%	36.1%	27.3%
	2 toothbrushes /day	60.4%	39.6%	59.3%	57.9%	57.1%	78.8%	61.7%	44.3%	55.5%
	3 toothbrushes /day	68.0%	32.0%	10.0%	10.2%	10.0%	15.2%	12.4%	8.2%	7.7%
	From time to time	40.5%	59.5%	9.3%	7.9%	5.0%	3.0%	4.7%	11.5%	9.6%
*p*		0.001 *		0.799			0.003 *			
Number of toothbrushes changed/year	1–3 times/year	49.8%	50.2%	57.1%	60.2%	62.1%	39.4%	63.2%	50.8%	62.7%
	4- 6 times/year	72.5%	27.5%	27.9%	25.5%	26.4%	54.5%	28.5%	23.0%	21.1%
	When needed	48.5%	51.5%	15.0%	14.4%	11.4%	6.1%	8.3%	26.2%	16.3%
*p*		0.000 *		0.870			0.000 *			
Reasons to visit a dentist	Check-up	54.6%	45.4%	28.6%	34.7%	34.3%	51.5%	38.3%	27.9%	26.3%
	Emergency	59.8%	40.2%	47.9%	38.4%	35.0%	33.3%	31.1%	47.5%	47.4%
	Treatment	54.0%	46.0%	21.4%	25.5%	27.9%	15.2%	25.9%	24.6%	25.8%
	I didn’t go to the dentist	10.0%	90.0%	2.1%	1.4%	2.9%		4.7%		0.5%
*p*		0.018 *		0.383			0.001 *			
Payment for dental services	Never been to a dentist	10.0%	90.0%	1.4%	1.9%	2.9%		4.7%		0.5%
	State insurance	62.1%	37.9%	88.6%	0.0%	0.0%	45.5%	2.1%	3.3%	49.3%
	Private insurance	64.0%	36.0%	0.0%	4.2%	11.4%		8.3%	8.2%	1.9%
	Self-funded	54.0%	46.0%	10.0%	94.0%	85.7%	54.5%	85.0%	88.5%	48.3%
*p*		0.009 *		0.000 *			0.000 *			
Preferred dental clinic	I don’t frequent any clinic	10.0%	90.0%	1.4%	1.9%	2.9%		4.7%		0.5%
	Private clinic	57.5%	42.5%	77.9%	78.7%	77.9%	66.7%	77.2%	70.5%	83.3%
	Government clinic	53.1%	46.9%	20.7%	19.4%	19.3%	33.3%	18.1%	29.5%	16.3%
*p*		0.010 *		0.930			0.002 *			
Self-rated oral health	Very good/good	55.1%	49.1%	50.7%	53.7%	52.1%	84.8%	58.0%	52.5%	42.1%
	Fair	29.0%	24.5%	22.9%	25.5%	33.6%	12.1%	29.5%	32.8%	25.4%
	Poor/very poor	15.9%	26.4%	26.4%	20.8%	14.3%	3.0%	12.4%	14.8%	32.5%
*p*		0.170		0.075			0.000 *			
Self-rated dental treatment need	Yes	37.0%	42.7%	38.6%	41.7%	37.1%	21.2%	39.9%	41.0%	41.6%
	No	63.0%	57.3%	61.4%	58.3%	62.9%	78.8%	60.1%	59.0%	58.4%
*p*		0.192		0.670			0.166			

* Statistically significant differences when *p* < 0.05 (ANOVA test).

**Table 4 medicina-59-00074-t004:** Multivariate regression between gender, occupation, monthly income (independent variables) and dependent variables that attest the access to medical services and hygiene attitudes.

		B	S.E.	*p*	OR	95% C.I.for EXP(β)	
						Lower	Upper
Number of toothbrushing/day							
	Gender	0.557	0.166	0.001 *	1.746	1.260	2.418
	Occupation	0.763	0.188	0.000 *	2.144	1.483	3.100
	Monthly income	0.387	0.204	0.059	1.472	0.986	2.197
Number of toothbrushes changed/year							
	Gender	0.999	0.193	0.000 *	2.714	1.859	3.962
	Occupation	0.451	0.209	0.031 *	1.570	1.042	2.367
	Monthly income	0.146	0.232	0.529	1.158	0.734	1.826
Reasons to visit the dentist							
	Gender	0.195	0.161	0.227	1.215	0.886	1.665
	Occupation	−0.594	0.182	0.001 *	0.552	0.386	0.790
	Monthly income	0.110	0.196	0.573	1.116	0.761	1.638
Payment for dental service							
	Gender	2.545	1.064	0.017 *	12.737	1.583	102.501
	Occupation	3.682	1.176	0.002 *	39.744	3.963	398.585
	Monthly income	−23.976	1564.422	0.988	0.000	0.000	0.000
Self-rated oral health							
	Gender	−0.334	0.158	0.035 *	0.716	0.525	0.977
	Occupation	−0.708	0.179	0.000 *	0.493	0.347	0.699
	Monthly income	−0.179	0.195	0.359	0.836	0.570	1.226
Self-rated dental treatment need							
	Gender	0.180	0.161	0.264	1.197	0.873	1.641
	Occupation	0.206	0.181	0.255	1.229	0.862	1.753
	Monthly income	0.050	0.200	0.801	1.052	0.711	1.556

* Statistically significant differences when *p* < 0.05.

## Data Availability

Not applicable.

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
