# Peer review of "Trends in Access to Oral Health Care among Adults from the N-E Region of Romania"

_medicina, 2022, doi:10.3390/medicina59010074_

Round 1
Reviewer 1 Report
This is a well written manuscript.
However one major change required.
1.The explanation for the results should be in discussion, not in results.For example why females are not visiting should be in discussion. Results should only be explaining the tables
2. Several grammatical mistakes identified.Please do extensive editing
Author Response
Dear Reviewer,
I would like to thank you once again for the precisely done review. It was a great experience to follow all the indications and comments that you gave us. We followed all your recommendations which increased the value of this manuscript and gave us additional, valuable knowledge.
Best regards,
Yours sincerely,
on behalf of the authors
BalcoÈ™ Carina
Response to Reviewer 1 Comments
1.The explanation for the results should be in discussion, not in results. For example why females are not visiting should be in discussion. Results should only be explaining the tables
R: I removed the comments from the results section as you suggested.
- Several grammatical mistakes identified. Please do extensive editing
R: The article was corrected by an authorized person.
Thank you very much for the guidance provided.
Please see the attachment.

Reviewer 2 Report
1. What is the significance of your study and its outcomes for the global scenario?
2. Since it is a questionnaire based study, how was it conducted restrospectively? Weren't participants enrolled sequentially to the study? If so, it would be a prospective study.
3. What was the sampling frame for the study, and how was it calculated and arrive at?
4. How were cases selected from the pre-existing dataset? Was their any form of randomization employed?
5. Is there any study reported from the regions of Romania with better socioeconomic indicators to compare with your study and prove that, poor dental health concern was only due to financial status?
Author Response
Dear Reviewer,
I would like to thank you once again for the precisely done review. It was a great experience to follow all the indications and comments that you gave us. We followed all your recommendations which increased the value of this manuscript and gave us additional, valuable knowledge.
Best regards,
Yours sincerely,
on behalf of the authors
BalcoÈ™ Carina
Response to Reviewer 2 Comments
1. What is the significance of your study and its outcomes for the global scenario?
R: This study, part of a doctoral research on the oral health of adults in the N-E region of Romania, wants to highlight the factors that influence access to dental medical services in a developing country, but in which the health system is underfunded, and the lack of prevention programs determines the adaptation of sanogenic behaviours to the current socio-economic situation. Through this research we want to provide data that can be compared with those of other regions, although there are no studies now, as well as with those of other countries. We collected the data on the socio-economic development of all regions from the documents drawn up by the Ministry of Finance or other european sources and not from studies. There is only one study carried out in Romania by region that analyzes the economic and dental health determinants.
- Since it is a questionnaire-based study, how was it conducted retrospectively? Weren't participants enrolled sequentially to the study? If so, it would be a prospective study.
R: We consider this study to be retrospective because the data were collected through the questionnaire during an oral health screening action that has been going on for 12 years in the N-E region of Romania, called the Prophylaxis Caravan. For this study, we used data from the database of this action registered in May 2022. The data analysis was carried out after obtaining approval from the Ethics Commission in October 2022.
- What was the sampling frame for the study, and how was it calculated and arrive at?
R: To an adult population of of 3.712.396 people, we applied the calculation formula with a confidence level of p = 95%, z = 1.96, as well a margin of error of 5% and the sample size was 385 adults. Of the total questionnaires completed at the last action, 696 were completed by adults, which is why we selected all the questionnaires completed by adults in 2022.
- How were cases selected from the pre-existing dataset? Was their any form of randomization employed?
R: We randomly selected the subjects more according to age than according to any other criterion.
- Is there any study reported from the regions of Romania with better socioeconomic indicators to compare with your study and prove that, poor dental health concern was only due to financial status?
R: There is only one study carried out in Romania by region that analyzes the economic and dental health determinants over the period 2001–2015. (Cigu, A.T.; Cigu, E. Exploring Dental Health and Its Economic Determinants in Romanian Regions. Healthcare 2022, 10, 2030. https://doi.org/10.3390/healthcare10102030)

Reviewer 3 Report
Introduction
Unnecessary information about income in Romania. Reword the entire paragraph. Instead of displaying the amount of income, you can create a scale based on which you will create a variable as social status.
Material i methods
OK
Results
The results are described in too much detail. There are also parts that are more up for discussion. This section needs to be shortened.
Univariate logistic regression was not performed. The p value no longer implies a certain significance, today the OD RATIO calculation prevails.
Discussion
The marked part in the discussion is too broad and mostly unnecessary. Rework.
Author Response
Dear Reviewer,
I would like to thank you once again for the precisely done review. It was a great experience to follow all the indications and comments that you gave us. We followed all your recommendations which increased the value of this manuscript and gave us additional, valuable knowledge.
Best regards,
Yours sincerely,
on behalf of the authors
BalcoÈ™ Carina
Response to Reviewer 3 Comments
Introduction
Unnecessary information about income in Romania. Reword the entire paragraph. Instead of displaying the amount of income, you can create a scale based on which you will create a variable as social status.
R: I rewrote the paragraph according to your instructions.
Results
The results are described in too much detail. There are also parts that are more up for discussion. This section needs to be shortened. Univariate logistic regression was not performed. The p value no longer implies a certain significance, today the OD RATIO calculation prevails.
- I tried to restructure the results. I have removed the comments that are better for discussions. Being a descriptive study, we did not use logistic regression as an analysis method.
Discussion
The marked part in the discussion is too broad and mostly unnecessary. Rework.
R: I have restructured the discussion part according to your recommendations.

Reviewer 4 Report
Dear Authors.
The manuscript is well written .However few minor corrections should be done in abstract .
Abstract :
1. kindly elaborate the material and methods in abstract
2. “In relation to occupation, there were significant differences for most of the variables evaluated.” Kindly elaborate
3. The Statement is not clear. Kindly rewrite.” Payment for dental services was made from state insurance for those with low monthly income and their own pocket in the case of higher monthly income.”
Overall the manuscript is written well. The author needs to elaborate the abstract.
Author Response
Dear Reviewer,
I would like to thank you once again for the precisely done review. It was a great experience to follow all the indications and comments that you gave us. We followed all your recommendations which increased the value of this manuscript and gave us additional, valuable knowledge.
Best regards,
Yours sincerely,
on behalf of the authors
BalcoÈ™ Carina
Response to Reviewer 4 Comments
Abstract :
- kindly elaborate the material and methods in abstract
R: I have added more information in the abstract, to the material and methods section
- “In relation to occupation, there were significant differences for most of the variables evaluated.” Kindly elaborate
R: I was referring to variables such as no. of brushes/day, no. of toothbrushes / year, reasons to visit a dentist, payment for dental services, preferred dental clinic, self-rated oral health(table 3)
- The Statement is not clear. Kindly rewrite.”Payment for dental services was made from state insurance for those with low monthly income and their own pocket in the case of higher monthly income.”
R: I have corrected the statement ‘’Dental services were paid for from state insurance for those with low monthly incomes and out-of-pocket for those with higher monthly incomes’’.
Thank you very much for the guidance.

Round 2
Reviewer 2 Report
Appreciate your responses to the review comments and the changes made to the manuscript.
Author Response
Thank you very much. Happy holidays.
Reviewer 3 Report
Univariate regression has not yet been performed. There are no parameters that stand out for significance (not based on ANOVA processing, but based on the calculation of OD RATIO.
Author Response
Dear Reviewer,
I tried to complete the statistical analysis with logistic regression and odd ratio between the variables analyzed in the study as you recommended. I hope the results are good. Thanks a lot for the guidance.
Happy holidays.
Sincerely,
Carina Balcos
